# Microbiome Analysis Reveals Biocontrol of *Aspergillus* and Mycotoxin Mitigation in Maize by the Growth-Promoting Fungal Endophyte *Colletotrichum tofieldiae* Ct0861

**DOI:** 10.3390/plants14213236

**Published:** 2025-10-22

**Authors:** Sandra Díaz-González, Carlos González-Sanz, Sara González-Bodí, Patricia Marín, Frédéric Brunner, Soledad Sacristán

**Affiliations:** 1Centro de Biotecnología y Genómica de Plantas (CBGP UPM-INIA/CSIC), Universidad Politécnica de Madrid (UPM)-Instituto Nacional de Investigación y Tecnología Agraria y Alimentaria–CSIC (INIA/CSIC), Campus Montegancedo, 28223 Pozuelo de Alarcón, Madrid, Spain; carlos.gonzalez.sanz@upm.es (C.G.-S.); sara.gonzalez.bodi@upm.es (S.G.-B.); 2Departamento de Biotecnología-Biología Vegetal, Escuela Técnica Superior de Ingeniería Agronómica, Alimentaria y de Biosistemas, Universidad Politécnica de Madrid (UPM), 28040 Madrid, Spain; 3PlantResponse Biotech, S.L. Centro de Empresas, Campus de Montegancedo, 28223 Pozuelo de Alarcón, Madrid, Spain

**Keywords:** plant microbiome, microbial diversity, maize, *Colletotrichum tofieldiae*, *Aspergillus*, mycotoxin, aflatoxin, bioinoculant, biocontrol, induced resistance

## Abstract

Maize (*Zea mays* L.) is a globally critical crop that faces numerous challenges, including contamination by mycotoxigenic fungi such as *Aspergillus* spp. The use of fungal endophytes as bioinoculants offers a sustainable strategy to improve plant resilience against biotic and abiotic stresses. Here, we evaluate the potential of *Colletotrichum tofieldiae* strain Ct0861 as a bioinoculant and its impact on maize-associated bacterial and fungal microbiomes. Field trials demonstrated that Ct0861 enhanced biomass and yield compared to controls, regardless of the application method (seed or foliar). Microbiome profiling showed that Ct0861 induced subtle, compartment-specific changes in microbial diversity and composition, while preserving the stability of core microbiome assemblages. Both microbiome data and qPCR quantification confirmed a significant reduction in *Aspergillus* spp. abundance in Ct0861-treated plants. Greenhouse assays corroborated these results: Ct0861 reduced *A. flavus* biomass by up to 90% and significantly lowered aflatoxin levels in infected grains. Dual-culture assays and the absence of Ct0861 in grain samples suggest an indirect biocontrol mechanism, potentially mediated by plant-induced resistance. This study provides the first evidence that Ct0861 acts as a biocontrol agent against mycotoxigenic *Aspergillus* spp. in maize. Beyond promoting plant growth, Ct0861 enhances food safety by reducing mycotoxin accumulation without disrupting the native microbiome, supporting its potential as a tool for sustainable crop protection.

## 1. Introduction

Maize (*Zea mays* L.) is a globally important crop, providing food, feed, and raw materials for various industries [1]. Ensuring its productivity and quality is critical to addressing global food security challenges, especially under increasing pressures from climate change, soil degradation, and pathogen prevalence [2,3]. Among these threats, mycotoxins—harmful secondary metabolites produced by fungi such as *Aspergillus* and *Fusarium* spp. [4]—pose a serious risk to maize safety and marketability [5]. These toxins not only compromise crop quality but also have severe implications for human and animal health, contributing to substantial economic losses and food insecurity, mostly in developing countries [6,7]. Therefore, strategies to enhance maize productivity must address both yield improvements and the mitigation of mycotoxin contamination to ensure sustainable and safe agricultural practices.

Recent advancements in agricultural biotechnology, particularly microbial inoculants, offer sustainable alternatives to enhance crop growth and resilience [8,9,10]. Among these, fungal endophytes have emerged as valuable candidates for developing novel microbial-based biocontrol or biostimulant products [11,12]. Fungal endophytes are a highly diverse group of fungi that naturally colonize internal plant tissues without causing disease symptoms for at least part of their life cycle [13]. These fungi can provide different benefits to their hosts, like a sort of Swiss knife with multiple tools for different situations [14,15,16]. A well-known example is *Serendipita indica*, which promotes vegetative growth, nutrient uptake, and plant yield across multiple hosts, including *Arabidopsis thaliana*, barley, maize, wheat, tomato, and cucumber, while also improving tolerance to biotic and abiotic stressors [17].

Microbial inoculants interact not only with host plants but also with the surrounding microbiota, potentially influencing the composition and diversity of plant-associated microbial communities [18,19,20]. While many inoculants establish beneficial interactions through diverse mechanisms, their effects on indigenous plant microbiomes are less studied [20]. Understanding these interactions is critical, as the plant microbiome plays a pivotal role in host fitness, nutrient cycling, and disease suppression [21,22].

The fungal endophyte *Colletotrichum tofieldiae* strain Ct0861 (Ct0861) was originally isolated from surface-disinfected leaves of asymptomatic *A. thaliana* plants from a natural population in Central Spain [23]. Ct0861 establishes mutualistic interactions with *A. thaliana*, promoting growth and fertility by facilitating phosphate translocation under phosphate-limiting conditions [24,25]. In a subsequent study, we demonstrated that Ct0861 successfully colonizes maize and tomato plants, promoting seedling growth and significantly increasing yield under standard grower conditions, and highlighting the agricultural potential of Ct0861 as a bioinoculant [26]. For maize, in particular, seed application produced a significant yield increase up to 22% [26].

In this study, we have investigated the effects of Ct0861 application on bacterial and fungal communities in various maize-associated compartments by using amplicon sequencing at two time points: one month after Ct0861 treatment and four months after sowing, at the end of the plant cycle. For that, we have tested two different Ct0861 application methods: seed coating and foliage spraying on one-month-old plants. In the first application method, the inoculant was applied to the seed with the aim that it colonizes and multiplies within the plant. In the second one, the application added larger amounts of inoculant to the foliage and surrounding soil. None of the two Ct0861 application methods showed a severe impact on the plant-associated microbiome, except for the grain compartment. Indeed, grains from Ct0861-treated plants exhibited a marked reduction in *Aspergillus* spp., a fungal genus associated with the production of harmful mycotoxins. Controlled *Aspergillus flavus* grain inoculation in greenhouse assays further confirmed these results, also showing reduced aflatoxin levels in the grains of Ct0861-treated plants. Thus, our study shows a novel potential activity of Ct0861 as a bioprotectant and underscores the dual benefits of this fungus in improving plant productivity and mitigating the presence of mycotoxigenic fungi. These results offer a promising sustainable solution for current challenged agroecosystems.

## 2. Materials and Methods

### 2.1. Fungal Inocula

The fungus *C. tofieldiae* strain Ct0861, originally isolated in the central Iberian Peninsula [23], was used to treat maize plants in both field and greenhouse experiments. The fungus *A. flavus* strain NRRL 6540, which produces aflatoxins B1 and B2 [27], was employed in controlled greenhouse infection assays.

To prepare the fungal inocula, potato–dextrose–agar (Difco^TM^ PDA; Benton, Dickinson and Company, Sparks Glencoe, MD, USA) plates were inoculated in the centre with a 3 µL drop of a conidia suspension of 10^6^ conidia/mL obtained from a 10^8^ conidia/mL stock stored at −80 °C in 2% skimmed milk (Sveltesse, Nestle^®^, Barcelona, Spain). Inoculated plates were incubated in a growth chamber at 24 °C under a light cycle of 14 h light/10 h darkness for Ct0861 and continuously in darkness for *A. flavus*.

### 2.2. Plant Material and Ct0861 Applications

The maize variety LG 34.90 (Limagrain Ibérica S.A., Navarra, Spain) was used in this study. Microbiome profiling examined four treatments: Ct0861 seed application, Ct0861 spray application, and their respective controls (Seed–Control, Seed–Ct0861, Spray–Control, Spray–Ct0861). The seed treatment involved applying 3 mL of a Ct0861–conidia suspension (3 × 10^6^ conidia/mL, equivalent to 2 × 10^4^ conidia/seed) or water (control) to 100 g of seeds (~430 seeds) in clean plastic bags. The bags were shaken vigorously to ensure even distribution of conidia on the seed surface, and the seeds were sown immediately after treatment [26]. For the spray application, maize seeds were sown simultaneously with the seed-treated seeds. At the V4–V5 growth stage (approximately one month after sowing), plants were treated with a Ct0861 spray (10^6^ conidia/plant, or 10^11^ conidia/ha) or water as a control.

### 2.3. Site Description, Irrigation and Fertilization

The study was conducted from May to October 2018 in the field of an experimental station located at Pilar de la Horadada, Alicante (37°51′46.0″ N–0°48′30.6″ W, 35 m above mean sea level) in the south-eastern part of the Iberian Peninsula, as described in [26] and [28] (Appendix A). According to Köppen–Geiger classification, the climate in this area is hot steppe (BSh) [29] with average annual temperature of 17.6 °C (ranging from 10.8 °C in January to 25.5 °C in August) and mean annual humidity of 71% [30]. Mean annual precipitation is 313 mm, with June, July, and August being especially dry months (average of 7, 2, and 7 mm, respectively [29]). The actual monthly accumulated precipitation in 2018 was: 2.6 mm in May (average: 0.08 mm/day), 7.3 mm in June (average: 0.4 mm/day), no precipitation events in July and 0.1 mm in August (average: 0.003 mm/day) [31].

Seeds were sown using a manual grain seeder, with a spacing of 18 cm between plants and 75 cm between rows (planting framework: 18 × 75 cm^2^). Each treatment had four plots consisting of three rows of 9 m and a total area of 20.25 m^2^. To minimize edge effects, border rows all along the field and buffer rows delimiting each plot were included. Due to the potential cross-contamination of control plants by Ct0861, plots containing control treatments, both seed and spray application, were separated a distance of 6 m from the Ct0861-inoculated plots.

Plants were watered daily during the night by a drip irrigation system and water requirements (see Appendix A) were calculated using long-term precipitation and evapotranspiration data from the closest climate station, located in San Javier (37°47′20″ N, 0°48′12″ O) [31]. Kc constant calculated with Cropwat v8.0 [32] was used to adapt the water requirements to the different developmental stages of the plant (Appendix A). Soil physical and chemical properties were analysed by Fitosoil Laboratorios S.L. (Murcia, Spain) in both control and Ct0861-treated plots to ensure soil homogeneity across treatments (Appendix A). No differences were found among the plots of the different treatments either before or after the experiment. Fertilization was adjusted to meet maize nutritional requirements, as described in [26].

### 2.4. Plant Growth and Yield Measurements

Stem height was recorded from 20 plants per plot (80 plants per treatment) randomly selected from the middle row of each plot at reproductive stage R4 (grain dough stage) [33]. Stem height was determined from the ground to the beginning of the male inflorescence (tassel). Root and shoot biomass were evaluated in four plants per plot (16 plants per treatment) randomly selected from lines 1 and 3 of each plot at reproductive stage R4. The stalk was cut 10 cm above ground and shoots were weighed fresh. The roots were lifted out of the ground with a shovel and washed with tap water to remove soil residues. Then, they were dried at room temperature until no change in weight was observed and weighed. The number of mature cobs per plant was recorded for 4 plants per plot (16 plants per treatment). To calculate mean cob weight per plant, mature cobs from 44 plants randomly selected from the central rows of the plot were collected and weighed together. Cobs were shelled, and the grains were weighed together to determine the mean plant yield for each plot. The weight of 1000 grains was determined twice for each plot. Grain humidity was assessed in order to normalize weights to 15.5% of water content. Yield in t/ha was calculated according to trial plant density (75,000 plants/ha). Randomization was achieved by assigning each plant a unique identifier, followed by using the formula RANDBETWEEN in Microsoft Excel as a random number generator to select plants for harvesting.

Plant growth and yield results from seed application were previously published in [26].

### 2.5. Sample Collection, Processing, and DNA Extraction for Microbiome Analysis

Sample collection, processing and analysis were done as in [28]. In order to have a representation of the whole trial area, we defined the plot as our sample unit. One maize plant of the central row of each plot (4 plants per treatment, 16 plants in total, Figure 1) was randomly selected. Random selection was done as described in Section 2.4. The sampling was done at two time points: (i) 1MPT: one month after treatment (approximately one month after sowing for seed application and two months after sowing for spray application) and (ii) 4MPS: approximately 4 months after sowing, near the end of the crop cycle (reproductive stage R4) [33]. Bulk soil, rhizosphere, root, leaf, and grain compartments were collected from each plant (*n* = 32 per compartment, except grain). Grains were only collected at 4MPS (*n* = 16). In total, 144 samples were processed as specified in Appendix A.

Total genomic DNA was extracted from bulk soil, rhizosphere, and grain samples with FastDNA^TM^ Spin Kit for Soil (MP Biomedicals, Solon, OH, USA) following the manufacturer’s instructions. DNA extraction from root and leaf tissues was conducted with DNeasy Plant Mini Kit (Qiagen, Hilden, Germany), following the manufacturer’s instructions with a minor adaptation in elution volumes (30 µL instead of 100 µL). Centrifugation steps were carried out at 4 °C. DNA quality was verified by agarose electrophoresis (1.5% agarose, Agarose D1 Medium EEO, Conda Pronadisa, Madrid, Spain). DNA quantification was carried out with Quant-iT^TM^ PicroGreen^TM^ dsDNA Assay Kit (Sigma-Aldrich, St. Louis, MO, USA) in Varioskan^TM^ LUX Multimode microplate reader (Thermo Scientific, Singapore).

### 2.6. Amplicon Generation and Sequencing

In total, 50 µL of each DNA extract was sent to Sequentia Biotech S.L. (Barcelona, Spain) for DNA normalization, amplicon generation, library preparation, and sequencing. Bacterial and fungal amplicons were obtained by PCR from DNA extracts of bulk soil, rhizosphere, root, leaf, and grain samples by using primer pair 515f/806r [34] for bacterial 16S rRNA gene and ITS1-F [35]/ITS2 [36] for fungal ITS gene (Table 1). Amplicon simultaneous sequencing was conducted on Illumina’s HiSeq2500 platform, generating 2 × 250 bp reads.

### 2.7. Read Data Processing and Taxonomic Assignment

The Divisive Amplicon Denoising Algorithm 2 (DADA2) (version 1.22.0) [37] was employed for the taxonomic identification of bacterial and fungal amplicons. The default parameters were used, with certain exceptions as specified in reference [37]. Initially, all raw reads were processed for adapter trimming using the ‘cutadapt’ tool (version 4.6) [38]. For the 16S rRNA gene sequencing data, standard filtering parameters were applied. These parameters allowed no uncalled bases, a maximum of 2 expected errors, and truncated reads at a quality score of 2 or less. In the case of the fungal ITS dataset, the same filtering approach was applied, with an additional parameter specifying a minimum read length of 50 bp. In both instances, learned errors were utilized to infer predicted error presence across all reads as a denoising measure.

Amplicon Sequence Variants (ASVs), where each ASV differs from the others by at least one nucleotide, were identified in each sample after sequence de-replication. Chimeric ASVs were determined with the consensus method. Subsequently, the ASVs were mapped to the SILVA v138 database [39] with a minimum bootstrap confidence level of 80 for bacterial assignment. The taxonomic assignment for ITS was performed against the UNITE v18.11.2018 database using the same bootstrap confidence level as above [40].

After DADA2 processing, the datasets retained the following number of reads: 64,729,520 (65.9% average among samples) for the 16S rRNA gene dataset, and 144,725,532 (76.9%) for the ITS dataset (Appendix A). The recovery of reads through both workflows was tracked for each step in each sample (Appendix A, respectively).

### 2.8. Bioinformatic Analysis

Downstream analyses were performed using RStudio software (version 4.2.2) and R (version 4.3.2). The resulting abundance table and taxonomic classification generated by DADA2 were imported into the phyloseq package (version 1.20.0) [41]. Plant reads were discarded. Sequence chimeras or artifacts were filtered out by removing ASVs without class or order taxonomic assignment. ASVs that were low represented with fewer than three reads and prevalent in less than 2% of the total samples were also discarded. Then, taxa counts were normalized using median sequencing depth. We obtained 7,618,901 (16S rRNA gene) and 72,832,867 (ITS) high-quality reads that were assigned to 11,471 bacterial ASVs and 2144 fungal ASVs (Appendix A).

#### 2.8.1. Diversity Analysis

Alpha-diversity (diversity within samples) analysis was carried out based on both the Chao1 richness estimator and the Shannon index at the ASV level with the function ‘estimate_richness’ (R package phyloseq version 1.20.0) [41]. Two-Sample Wilcoxon Test (*p* < 0.05) was conducted by the implementation of ‘wilcox.test’ function (R package stats version 4.4.1) [42] for the comparison between treatments. Beta-diversity analysis was carried out by calculating Bray–Curtis dissimilarity at the ASV level by using functions ‘distance’ and ‘ordinate’ (R package phyloseq version 1.20.0) [41]. Permutational multivariate analyses of variance (PERMANOVA) were performed using the ‘adonis2′ function implemented in the vegan package (version 2.6.4) [43]. Plots were generated with the ‘ggplot’ function (R package ggplot2 version 3.5.1) [44].

#### 2.8.2. Taxonomic Composition

R function ‘ps_venn’ (R_package MicEco version 0.9.19) [45] was used to extract the list of common and exclusive ASVs for controls and Ct0861-treated plants, while ‘venn.diagram’ (R package VennDiagram version 1.7.3) [46] was used for plotting. Taxa composition was calculated at the genus level. ASVs were merged to the genus level to determine relative abundance (Appendix A). ANOVA-Like Differential Expression analysis (ALDEx2) for the comparison of control and Ct0861 samples was conducted for the different datasets with the raw counts at de genus level using the function ‘aldex’ from the R package ‘ALDEx2’ (version 1.36.0) [47,48,49].

#### 2.8.3. *Aspergillus* spp. Abundance

Quantification of *Aspergillus* spp. in grain samples was conducted by filtering the taxonomy matrix associated with count data of ASVs assigned to that genus. Two-Sample Wilcoxon Test (*p* < 0.05) was conducted by the implementation of ‘wilcox.test’ function (R package stats version 4.4.1) [42] for the comparison between treatments.

### 2.9. Ct0861 and Aspergillus Biomass Quantification in Grain Samples

Ct0861, *A. flavus* and *Aspergillus niger* biomass in the grain were quantified by real-time qPCR both in field samples and in samples from greenhouse *A. flavus* infection assays. Grain samples were ground in liquid nitrogen and DNA was extracted following the CTAB extraction method [50]. Samples were diluted to 20 ng/µL and fungal biomass in the maize grains was quantified by real-time qPCR SYBR green reaction using LightCycler^®^ 480 System (Roche Diagnostics International Ltd., Rotkreuz, Switzerland) with the following cycling conditions: 95 °C 5 min, followed by 40 cycles of 95 °C 10 s, 60 °C 10 s and 72 °C 10 s. The primers used for quantification of *C. tofieldiae* Ct0861, *A. flavus* and *A. niger* are specified in Table 2.

### 2.10. Controlled Greenhouse Infection Assays with A. flavus in Maize Plants

Greenhouse infection assays of maize grains with *A. flavus* strain NRRL 6540 were performed following the method described by [53], with slight modifications. Maize seeds LG 34.90 (Limagrain Ibérica S.A., Navarra, Spain) were directly sown in 2.5 L containers filled with peat-vermiculite 3:1. For Ct0861 treatment, one week after sowing, plants were inoculated with 10 mL of a Ct0861 suspension at 10^5^ conidia/mL (10^6^ conidia/plant) or water for controls. After one month, plants were transplanted into larger containers (40 L) and reinoculated with the same dose of Ct0861 (10^6^ conidia/plant) or water. When the plants reached the reproductive stage, they were hand-pollinated and, 15 days post-pollination (grain fill stage R2-Blister) [33], they were inoculated with *A. flavus* strain NRRL 6540. Plants were grown at temperatures between 19 °C and 25 °C.

For *A. flavus* inoculation, a conidial suspension (1 × 10^6^ conidia/mL) was prepared in 0.05% (*v*/*v*) Triton X-100 from a 7–10-day-old *A. flavus* culture grown on PDA as previously described (Section 3.1). Each cob was injected transversely at four points along the middle with 1 mL of the suspension, while water containing 0.05% (*v*/*v*) Triton X-100 was used as a control. Cobs were collected at 5 (grain fill stage R3-Milk) [33] and 21 (grain fill stage R5-Dent) [33] days post-inoculation (dpi), shelled, and grains immediately frozen in liquid nitrogen.

Infected grains from control untreated plants and Ct0861-treated plants were ground in liquid nitrogen using a mortar for subsequent analysis of *A. flavus* biomass and mycotoxin content. Fungal biomass was quantified as described above (Section 2.9) for grains from three plants per treatment at 5 dpi and 5 plants per treatment at 21 dpi. Mycotoxin levels—including aflatoxins (B1, B2, G1, G2), deoxynivalenol, fumonisins (B1 + B2), ochratoxin A, patulin, zearalenone, and T-2/HT-2 toxins—were quantified in grains collected at 21 dpi from five plants of each treatment. Analysis was performed by Fitosoil Laboratorios S.L. (Murcia, Spain) using liquid chromatography coupled with tandem mass spectrometry (LC-MS/MS).

### 2.11. Ct0861-A. flavus Dual Culture Bioassay

Ct0861 was tested for antagonism against *A. flavus* strain NRRL 6540 following the protocol described by [54], with few modifications. Mycelium plugs of 0.5 cm diameter were cut from the active growing edge of 7-day-old fungal cultures and placed at the edges of PDA plates with a distance of 6 cm. In test plates, Ct0861 and *A. flavus* plugs were placed facing each other. In control plates, two plugs of *A. flavus* were used. All plates were incubated in the dark at 24 °C for 7 days. Radial fungal colony growth was measured daily. The experiment was carried out with five replicates (five control and test plates).

### 2.12. Statistical Analyses

Statistical analyses were performed in RStudio software (version 4.2.2). In order to check normality and homogeneity of variances assumptions, both the Shapiro–Wilk Normality Test and the Bartlett test of Homogeneity of Variance were performed using ‘shapiro.test’ and ‘bartlett.test’ functions, respectively (R package stats version 4.4.1) [42]. Comparisons for data fitting a normal distribution were analysed by the Welch Two-sample *t*-test with ‘*t*.test’ (*p* < 0.05) function (R package stats version 4.4.1) [42]. Comparisons not meeting normality assumption were first checked for homogeneity of variances with Fligner-Killeen test by running ‘fligner.test’ function (R package stats version 4.4.1) [42] and then a Two-Sample Wilcoxon-Mann–Whitney Test (*p* < 0.05) was conducted by the implementation of ‘wilcox_test’ function (R package coin version 1.4.3) [55,56].

## 3. Results

### 3.1. Ct0861 Seed and Spray Applications Have Comparable Plant Growth and Yield Promotion Effects in Field Trials

In this work, we have analysed Ct0861’s effect on plant growth and microbiome by using two different application methods: seed application, in which seeds have been soaked in a Ct0861 spore suspension before sowing, and foliage spray application, where one-month-old plants have been sprayed with a spore suspension. Results of plant growth and yield effects of Ct0861 applied with each method are summarized in Table 3, including previously published data for seed applications [26].

Plant emergence and survival measured at 30 days post-sowing were between 95% and 98% and did not differ between Ct0861 treatments and controls. For each treatment, the growth of maize plants was assessed by measuring the stem height and root and shoot weights at phenological stage R4. Plants treated with Ct0861 either on the seed or as a spray were significantly taller than non-inoculated plants (seed: *p* < 0.001; spray: *p* < 0.001; Table 3). Ct0861-treated plants showed an increase in shoot biomass (*p* = 0.008; Table 3) and root dry weight (*p* < 0.001; Table 3), being both significant in the spray application.

Yield parameters were evaluated at the end of the plant cycle (Table 3). Ct0861-treated plants showed an increase in the number of cobs per plant (5% in seed application and 16% in spray application), though these differences were not statistically significant. Cob weight per plant increased by 20% in the seed application (non-significant) and by 21% in the spray application, which was statistically significant (*p* = 0.037; Table 3). Similarly, 1000-grain weight increased by 4% (non-significant) in seed-treated plants and by 6% in spray-treated plants, the latter being statistically significant (*p* = 0.008; Table 3).

The yield of control plants was consistent with the expected values provided by the seed breeder and comparable to the average yield of irrigated maize in Spain in 2018 (12.3 t/ha) [57]. This was 1.5 t/ha higher than the average yield in Alicante province (11 t/ha) in that year. Ct0861 seed-treated plants produced a significantly higher yield than controls, with a 22% increase, corresponding to 2.9 t/ha and 4.2 t/ha more than the national and provincial averages, respectively. Ct0861 spray-treated plants also exhibited a yield increase of 21%, although not statistically significant, corresponding to 2.8 t/ha and 4.1 t/ha above the national and provincial averages.

Growth and yield parameters were comparable between seed and spray applications, with no significant differences attributed to the method of Ct0861 application. Overall, the results show that Ct0861 treatment enhances biomass and yield compared to controls, irrespective of the application method.

### 3.2. Ct0861 Treatment Decreased Alpha-Diversity of Fungal Communities in the Grain

Alpha-diversity indices, Chao1 and Shannon, were calculated to assess the effects of Ct0861 treatment across various compartments (Figure 2 and Appendix A). For the bacterial community, Ct0861 treatment had no significant effect on Chao1 or Shannon indices in any compartment. Similarly, fungal communities showed no significant changes, except in the grain. In grains from Ct0861-treated plants, both Chao1 (*p* = 0.003) and Shannon (*p* = 0.007) indices were significantly reduced.

No significant differences were found when comparing each application method separately. When comparing with control treatments, just a slight effect was found for Ct0861 spray application for Shannon indices of fungi in the rhizosphere and bacteria in the root, being both increased. However, a strong reduction effect on both Chao1 and Shannon indices for fungal grain assemblage was also found for spray application, consistent with the results observed with the aggregated data of both types of Ct0861 applications (Appendix A).

When disaggregating data by sampling time point within each application mode, Ct0861 treatment occasionally caused punctual changes in some samples, but no consistent trends were observed (Appendix A). For instance, Chao1 was significantly increased in spray-Ct0861-treated bulk soil at 1MPT for bacteria and in the rhizosphere at 4MPS for fungi. Conversely, it was significantly decreased for fungi in the seed-Ct0861-treated leaf at 1MPT. Similarly, the Shannon index was higher in the roots of spray-Ct0861-treated plants at 1MPT for bacteria but lower in the bulk soil of spray-Ct0861-treated plants at 4MPS. These results indicate that Ct0861 treatment did not have a strong general effect on bacterial and fungal alpha-diversity in maize-associated microbiomes, with the notable exception of reduced fungal diversity in grains, particularly in spray-treated plants (Figure 2 and Appendix A).

### 3.3. Ct0861 Application Exerts an Influence on Microbial Beta-Diversity, Mainly in the Soil Compartments and the Grain

The impact of Ct0861 application on beta-diversity (diversity among samples) was analysed using Bray–Curtis dissimilarity and PERMANOVA (Appendix A). As expected, compartment was the primary driver of microbiome differentiation (Bacteria: *p* < 0.001, R^2^ = 0.386; Fungi: *p* < 0.001, R^2^ = 0.488), followed by sampling time point (Bacteria: *p* < 0.001, R^2^ = 0.049; Fungi: *p* < 0.001, R^2^ = 0.067). In contrast, neither the type of application (Bacteria: *p* = 0.125, R^2^ = 0.010; Fungi: *p* = 0.783, R^2^ = 0.004) nor the treatment itself (Bacteria: *p* = 0.352, R^2^ = 0.007; Fungi: *p* = 0.132, R^2^ = 0.010) had a significant effect on the aggregated dataset.

Given the strong influence of compartment and sampling time point (Appendix A), PERMANOVA analysis was subsequently applied to datasets filtered by these factors (Figure 3). Ct0861 significantly affected both bacterial and fungal microbiome compositions in the soil at 1MPT (Bacteria: *p* = 0.014, R^2^ = 0.14; Fungi: *p* < 0.001, R^2^ = 0.17) and at 4MPS (Bacteria: *p* = 0.007, R^2^ = 0.14; Fungi: *p* = 0.002, R^2^ = 0.14), as well as in the rhizosphere at 4MPS (Bacteria: *p* = 0.005, R^2^ = 0.13; Fungi: *p* = 0.006, R^2^ = 0.14). The fungal microbiome in the roots was also significantly impacted at 1MPT (*p* = 0.002, R^2^ = 0.13). Notably, the greatest effect of Ct0861 treatment was observed in the fungal microbiome of the grain, where it explained 31.2% of the variance (*p* < 0.001).

When disaggregating the data by application mode, different significant effects were observed depending on compartment and time of sampling, with no general trend. For example, Ct0861 seed treatment significantly affected beta-diversity in the bacterial assemblage in the rhizosphere at 1MPT and leaf at 4MPS, and both bacterial and fungal assemblages at root at 4MPS. Ct0861 spray treatment significantly affected beta-diversity in the bacterial assemblage in the soil at 4MPS and root at 1MPT and both bacterial and fungal assemblages at leaf at 1MPT. Just soil at 1MPT and, notably, grain at 4MPS, showed a consistent differentiation for both bacterial and fungal assemblages, explaining up to 50% of the variance in the case of the fungal assemblage in the grain of Ct0861 spray-treated plants.

In summary, both seed and spray Ct0861 treatments have a certain effect on the assemblage differentiation of soil samples at 1MPT. This differentiation is attenuated in other compartments, except in the grain (Figure 3).

### 3.4. Ct0861-Treated Plants Showed Reduced Abundances of Aspergillus spp. in the Grain

To explore the taxonomical differentiation of microbial assemblages in Ct0861-treated plants, we first studied the proportion of taxa common or exclusive in control and Ct0861-treated plants. Venn diagrams show a core microbiome of 9782 bacterial ASVs and 1955 fungal ASVs present in both control and Ct0861-treated plants, which account for 85% and 91% of full bacterial and fungal assemblages, respectively (Appendix A). Ct0861-treated exclusive taxa represented only a small proportion of ASVs: 599 (5%) in bacteria and 99 (4.6%) in fungi.

To determine the differential abundance of bacterial and fungal taxa between controls and Ct0861, ALDEx2 analysis was conducted across the different dataset combinations at the genus level (Figure 4, Appendix A). This analysis identified 40 bacterial and 26 fungal genera as differentially abundant between Ct0861-treated and control plants. Among bacteria, most differentially abundant genera were observed at 1MPT in the spray treatment. Conversely, most differentially abundant fungal genera were associated with seed-treated plants.

Differentially abundant genera were predominantly detected in soil compartments (bulk soil and rhizosphere) of Ct0861-treated plants, under different conditions of sampling time and application method (Figure 4, Appendix A). Within plant compartments, only a few taxa showed differential depletion: seven bacterial genera in the roots, one bacterial and three fungal genera in the leaves, and one bacterial and three fungal genera in the grains. In all cases, these changes reflected a depletion effect. Of particular interest, the fungal genus *Aspergillus* was significantly reduced in grains of Ct0861-treated plants compared to controls, both in seed and spray application (Figure 4E and Appendix A).

The genus *Aspergillus* groups various well-known species of mycotoxin-producing fungi [4,58,59]. This finding prompted further investigation into how Ct0861 treatments affect the relative abundance of *Aspergillus* spp. in the grain. Results revealed a significant reduction in the relative abundance (*p* < 0.001) of the genus *Aspergillus* in grain samples from Ct0861-treated plants (Figure 5A). While the mean relative abundance reached 26.4% in grains from control plants, *Aspergillus* spp. represented just 1.7% in grains from Ct0861-treated plants (*p* < 0.001). As shown in Figure 5B, the effect was similar with both Ct0861 application methods, with reductions of 26.7% (*p* = 0.029) and 22.5% (*p* = 0.029) of *Aspergillus* spp. mean relative abundance in the seed and spray treatments, respectively.

In order to validate the results obtained by amplicon sequencing, we conducted real-time qPCR quantification of *Aspergillus* biomass in the maize grains obtained from the plants of our experiment. Given the difficulties in obtaining generic primers for the whole genus, we decided to base our analysis on two important mycotoxigenic species, *A. flavus* and *A. niger*, using specific primers. *A. flavus* is one of the most important mycotoxin producers, especially because of the production of aflatoxin [58]. *A. niger* is a common contaminant in maize, producer of the mycotoxins fumonisins B1, B2, and B4 [60,61,62,63], with some strains that can also synthesize ochratoxin A [4,61,63]. Biomass quantification by real-time qPCR confirmed the results obtained in the microbiome analysis for *Aspergillus* genus (Figure 5C,D). *A. flavus* was reduced in the grains of plants treated with Ct0861, although only spray application was significant (*p* = 0.039). Although *A. niger* biomass was lower in the samples, we observed a significant reduction in the case of Ct0861-treated plants, in both seed (*p* = 0.024) and spray (*p* = 0.001) applications. Therefore, qPCR analyses confirmed that the application with Ct0861 reduced the biomass of *Aspergillus* spp. in the grain.

### 3.5. Ct0861 Treatment Reduces the Growth of Artificially Inoculated A. flavus and Aflatoxins Levels in Maize Grains

To confirm the protective effect of Ct0861 against *Aspergillus* spp. observed in microbiome samples, we conducted greenhouse experiments where we controlled the infection by *A. flavus* in the grains of maize plants. Maize plants were treated with Ct0861 at one and four weeks after sowing. Fifteen days after hand-pollination, the cobs were inoculated with *A. flavus* strain NRRL 6540, which produces aflatoxins [27]. Grains of Ct0861-treated and control plants were collected at 5 and 21 dpi to determine *A. flavus* biomass. Five days after the inoculation with *A. flavus*, the grains from Ct0861-treated plants showed a significant reduction of 90% (*p* = 0.019) of *A. flavus* biomass in comparison to control (Figure 6B). This effect was also observed at 21 dpi, with a significant decrease of 83% (*p* = 0.010, Figure 6A,B). These results were confirmed by two independent greenhouse experiments.

The levels of various mycotoxins were measured in grains collected at 21 dpi with *A. flavus*. Among the analysed mycotoxins, only aflatoxins B1 and B2 were detected in the samples. The results revealed a notable reduction in aflatoxin levels in grains from plants treated with Ct0861 (Figure 6C,D). In the control group, the mean concentrations of aflatoxin B1 and B2 were 72 µg/kg and 1.18 µg/kg, respectively. In contrast, grains from Ct0861-treated plants showed significantly lower levels, with aflatoxin B1 being reduced to 0.46 µg/kg, and aflatoxin B2 being undetectable.

The colonization level by Ct0861 was evaluated by real-time qPCR in the roots of control and Ct0861-treated plants. Ct0861 was detected in all the treated plants, while it was not detected in the controls.

### 3.6. Ct0861 Does Not Have a Direct Antagonistic Effect on A. flavus Growth

In order to check if the effect of Ct0861 was due to direct competence, antibiosis or predation on *A. flavus*, we used two approaches: (i) a dual culture bioassay with *A. flavus* strain NRRL 6540 and (ii) detection of Ct0861 by real-time qPCR in grain samples both from the microbiome study and the controlled infection assays. Regarding the dual culture bioassay, we did not find any reduction in the growth of *A. flavus* (Appendix A) after 7 days of co-cultivation with Ct0861. In fact, control plates where two plugs of *A. flavus* were faced showed a higher growth inhibition than test plates where *A. flavus* was faced against Ct0861. Ct0861 DNA was not detected in the grains, either in the samples used for microbiome amplicon sequencing or in those from controlled infection assays. These results indicate that the protection against *A. flavus* observed in Ct0861-treated plants is not likely due to a direct fungal interaction.

## 4. Discussion

### 4.1. Ct0861 Enhances Maize Growth and Yield Irrespective of the Application Method

This study validated the efficacy of Ct0861 treatment as a spray application for enhancing maize growth and yield, complementing previously reported results as a seed application [26]. Both application methods significantly improved plant growth, while yield increases reached 21–22% compared to controls (Table 3). The differences in both types of application relies in (i) the amount of applied inoculum (2 × 10^4^ conidia/seed in the seed application vs. 10^6^ conidia per plant in the spray application); (ii) the moment of application (at sowing in the seed application vs. one month after sowing in the spray application); and (iii) in the place of application (limited to the seed and surrounding soil in the seed application vs. the whole plant foliage and the soil below in the spray application). The general use of seeds treated with fungicide in conventional agriculture hinders the potential application of Ct0861 as a seed treatment product. Our findings demonstrate that spray application offers an effective alternative, highlighting the potential of Ct0861 as a valuable and sustainable technology to optimize maize productivity under real-world agricultural conditions.

### 4.2. Ct0861 Induces Subtle Compartment-Specific Shifts in the Maize Microbiome While Preserving Its General Structure

The introduction of high amounts of an exogenous microorganism, along with its positive effects on plant fitness and productivity, could impact the composition of the plant microbial community. To investigate whether Ct0861 treatment influences the plant microbiome, we analysed bacterial and fungal communities in bulk soil, rhizosphere, roots, leaves, and grains of treated and control maize plants using amplicon sequencing. Our findings indicate that Ct0861 treatment had overall slight compartment-specific effects on microbiome diversity in maize (Figure 2 and Figure 3). Beta-diversity analyses revealed that bacterial and fungal community composition was primarily driven by compartment and sampling time point (Figure 3 and Appendix A), as expected from other studies in the same site [28] or other places and plant species [64,65]. Nonetheless, Ct0861-treatment exerted a significant and localized effect on the microbiome structure of soil, rhizosphere and, more strongly, the grains. The differences in the application method, with spray application adding larger amounts of inoculant and being less localized than seed application, could also have different impacts on the plant-associated microbiome. However, no significant differences were found due to the application method, being the main effects observed on soil and grain compartments fully consistent between seed and spray application.

Our taxonomic analysis revealed that Ct0861 treatment had a modest impact on the composition of bacterial and fungal communities (Appendix A). The core microbiome shared between treated and control plants remained largely stable, representing the majority of bacterial and fungal ASVs. Differential abundance analysis revealed few shifts in bacterial and fungal communities, particularly within soil compartments (bulk soil and rhizosphere, Figure 4 and Appendix A). These shifts predominantly involved depletion effects. Plant-associated compartments displayed fewer changes, with limited differential depletion observed in roots, leaves, and grains (Figure 4C– E). Notably, fungi of the genus *Aspergillus* were significantly reduced in grains of Ct0861-treated plants (Figure 4E and Figure 5 and Appendix A). This reduction was consistent across both Ct0861 application methods, though it was more pronounced with the spray application (Figure 4E and Figure 5C).

While Ct0861 improved plant growth and yield, these benefits occurred without severe microbiome disruptions, reflecting subtle, yet meaningful, changes in microbial diversity and composition. This aligns with previous studies reporting varied microbiome responses to microbial inoculants. For instance, He et al. 2021 [66] observed that a treatment with dark-septate endophytes altered microbial diversity in a medicinal plant depending on watering conditions, compartment, and inoculant species. Similarly, Yurgel et al. 2022 [67] documented short-term diversity shifts in soil, rhizosphere, and root microbiomes of *Lonicera caerulea*, with niche-specific patterns. In strawberries, Sylla et al. (2013) [68] found that biocontrol products like *Bacillus amyloliquefaciens* and *Beauveria bassiana* had minimal impact on phyllosphere microbiota, although *Trichoderma harzianum* altered fungal community composition in treated leaves. *Epichlöe* endophytes also yield variable results, with some studies reporting increased rhizosphere bacterial diversity [69,70], while others noted increased diversity and richness in root-associated and rhizosphere fungal communities but no significant effect—or even reduced diversity—on bacterial communities [71,72]. Overall, our findings and prior research suggest that plant fitness improvements might not result from strong microbiome alterations but from nuanced interactions between the inoculant and plant-associated microbial communities.

### 4.3. Ct0861 Treatment Reduces Aspergillus spp. and Mycotoxin Contamination in Maize Grains, Likely Through Induced Resistance

Still, subtle changes in microbial communities, such as shifts in the abundance of specific taxa, can provide significant benefits to the plants. Particularly, our study demonstrated that Ct0861 treatment reduced the abundance of mycotoxigenic *Aspergillus* species in maize grains, resulting in lower mycotoxin levels, as confirmed through controlled infection experiments with *A. flavus* (Figure 6). *Aspergillus* is a fungal genus of particular concern due to its production of harmful mycotoxins [4,73]. Mycotoxins are low-molecular-weight toxic compounds produced during fungal secondary metabolism, posing serious health risks to humans and animals, especially in developing countries [6,73]. Beyond health hazards, mycotoxin contamination causes substantial economic losses in agricultural commodities worldwide. Consequently, reducing mycotoxin contamination in agricultural and food systems is a critical global objective [74].

Biological control agents have proven to be effective pre-harvest strategies for preventing the growth of toxigenic fungal species and the contamination of agricultural commodities [75]. For instance, *Saccharomyces cerevisiae* has shown strong potential to inhibit the growth of *A. flavus* and *A. parasiticus* through competitive exclusion [76]. Similarly, fast-growing *Trichoderma* isolates have effectively restricted the expansion of *A. flavus* colonies [77,78]. In addition, several bacterial and yeast isolates capable of synthesizing antifungal secondary metabolites, mostly volatile organic compounds (VOCs), have exhibited activity against various mycotoxigenic *Aspergillus* species, including *A. flavus* [79,80,81], *A. ochraceus* [82], and *A. carbonarius* [82,83]. The *in situ* production of lytic enzymes has also shown promise in limiting the growth of *Aspergillus* species. For example, strains of the yeast *Aureobasidium pullulans* produced chitinase and β-1,3-glucanase after five days of cultivation with *A. flavus* cell wall preparations [84]. However, the majority of these findings come from *in vitro* experiments, and their effectiveness under real agricultural conditions remains largely untested. Currently, the most widely adopted biocontrol strategy in agricultural fields involves the use of non-toxigenic strains of *Aspergillus*, primarily *A. flavus* [85,86,87,88]. These strains have demonstrated strong competitive performance against toxigenic strains in crops such as maize, cotton, peanuts, pistachios and sorghum [85,88,89]. These examples highlight direct effects of biocontrol organisms—such as competition, toxicity or displacement—in suppressing pathogen growth.

Interestingly, in our experiments, Ct0861 was not detected in the grains of treated plants via specific real-time qPCR, and dual-culture bioassays showed no direct effects on the growth of *A. flavus* (Appendix A). This suggests that the reduction of *Aspergillus* spp. in grains from Ct0861-inoculated plants is not due to direct mechanisms, such as competition or antibiosis. Instead, indirect mechanisms may be at play, such as the activation of induced resistance (IR) pathways. IR, often activated by beneficial microbes such as plant growth-promoting rhizobacteria (PGPR) or endophytes, typically depends on jasmonic acid (JA) and ethylene (ET) signalling pathways [90,91]. Although microbially IR to *Aspergillus* remains largely unexplored, some studies provide relevant insights. For instance, the rhizobacterium *Klebsiella* sp. MBE02 induced resistance in peanut plants against *Aspergillus* infection. RNA sequencing revealed upregulation of genes involved in JA, ET, and pathogen-defense signalling pathways. Remarkably, MBE02 treatment protected peanut plants from *Aspergillus* infection in both controlled and field environments [92]. Additionally, IR can be mediated by salicylic acid (SA), involving the accumulation of pathogenesis-related (PR) proteins with antifungal properties [91]. Foliar application of *A. flavus* and *A. parasiticus* culture filtrates on maize plants has been shown to trigger IR, enhancing defense-related enzyme activities, PR protein production, and plant growth [93]. *PR2* and *PR5*, marker genes of the SA-dependent defense pathways, might also be involved in the *Paenibacillus alvei* K165-mediated protection against *A. carbonarius* in grapes [94]. There is evidence that Ct0861 in plant roots can trigger immune pathways in distal tissues. For example, Ct0861 was shown to activate immune responses in *Arabidopsis* leaves, relying on indole glucosinolate metabolism, as well as JA, ET and SA signalling [95].

Ct0861 treatment may also influence the expression of specific genes related to maize resistance to *Aspergillus* spp. Although maize resistance to *A. flavus* and aflatoxin contamination is a complex trait involving multiple mechanisms that are not yet fully understood [53,96,97], prior research highlights several key plant genes and proteins involved. For example, seven chitinase genes have been identified with alleles associated with increased resistance to aflatoxin accumulation and *A. flavus* infection in field-grown maize [98]. Elevated β-1,3-glucanase activity in maize grains has also been linked to reduced *A. flavus* infection in resistant maize varieties [99]. Other resistance-related endosperm proteins include globulin-2, late embryogenesis abundant proteins (LEA3 and LEA14), a stress-related peroxiredoxin antioxidant (PER1), heat-shock proteins (HSP17.2), a cold-regulated protein (COR), and an antifungal trypsin-inhibitor protein (TI) [100]. Furthermore, some studies suggest that genes involved in fatty acid biosynthesis and antibiotic production play key roles in resistance to *A. flavus* [53]. Transcription factors induced in response to infection also differ between resistant and susceptible grains [101]. These studies suggest that resistance mechanisms involve intricate gene regulatory networks and interactions with environmental factors [53,96], and the implication of Ct0861 in this complex interaction delves further investigation.

Despite the unknown mode of action, the observed reduction in toxigenic fungi in grains represents the first protective effect shown for Ct0861, a significant finding that opens new avenues for research and potential applications. Further studies are needed to elucidate the mechanisms underlying this interaction and to explore the full potential of Ct0861 in agricultural contexts.

## 5. Conclusions

In summary, this study underscores the agricultural potential of Ct0861 as a versatile and flexible bioinoculant that enhances plant growth and yield without disrupting plant microbial diversity. Notably, grains from Ct0861-treated plants exhibited a marked reduction in *Aspergillus* spp., a key toxigenic fungal genus linked to post-harvest diseases. Controlled infection assays with *A. flavus* further validated these findings, showing reduced aflatoxin content in the grains of Ct0861-treated plants. This represents the first evidence of Ct0861 biocontrol potential. While the results suggest an indirect mechanism, possibly through the induction of plant resistance, further research is essential to unravel the underlying processes and explore their broader applicability across crops and conditions.

## Figures and Tables

**Figure 1 plants-14-03236-f001:**
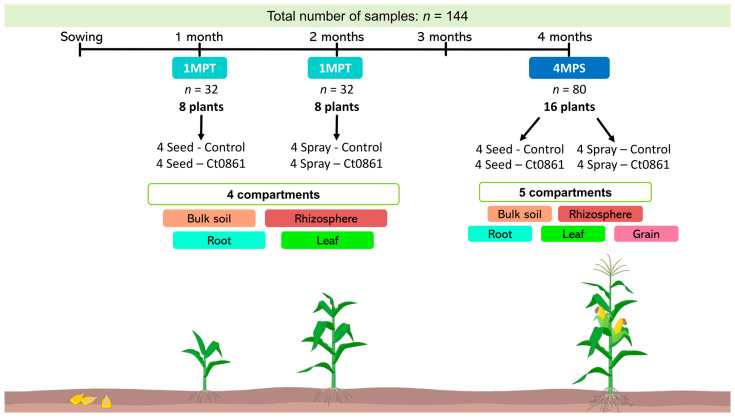
Sampling design and timeline of the experiment. For each treatment, one maize plant from the central row of each plot was randomly selected (4 plants per treatment; 16 plants total). Sampling was performed at two time points: (i) 1MPT (one month after treatment: 1 month after sowing for seed application and 2 months for spray application) and (ii) 4MPS (4 months after sowing, reproductive stage R4). Bulk soil, rhizosphere, root, leaf and grain samples were collected from each plant (*n* = 32 per compartment, except grains). Grains were sampled only at 4MPS (*n* = 16). In total, 144 samples were analyzed.

**Figure 2 plants-14-03236-f002:**
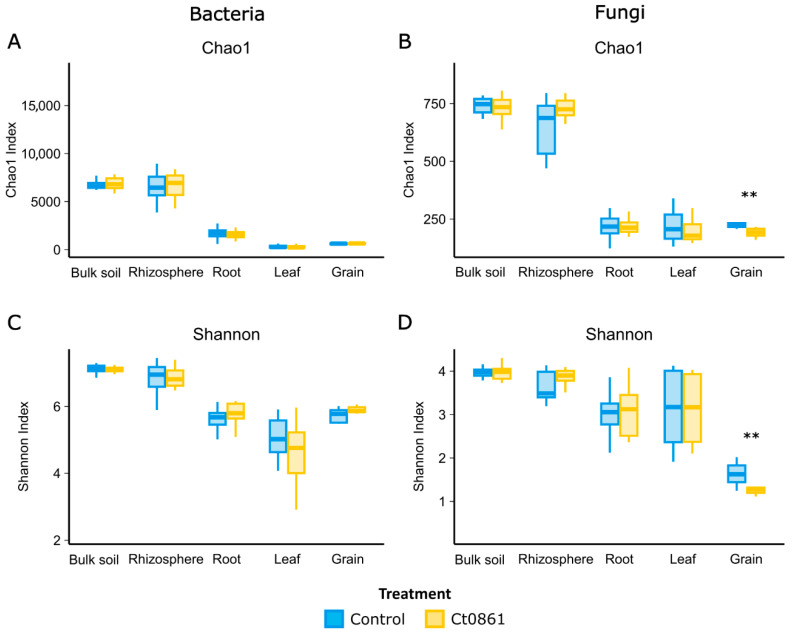
Alpha-diversity of maize-associated microbiome treated or not with Ct0861. Chao1 (**A**,**B**) Shannon indexes (**C**,**D**) of bacterial and fungal communities in different compartments (bulk soil, rhizosphere, root, leaf and grain) treated or not with Ct0861. Internal line in box-plot boxes indicates the median or second quartile (Q2), upper line of the boxes indicates the third quartile (Q3) and lower line the first quartile (Q1) of the data. Asterisks denote statistically significant differences in non-parametric Wilcoxon-Mann–Whitney Test (*p* < 0.01 **).

**Figure 3 plants-14-03236-f003:**
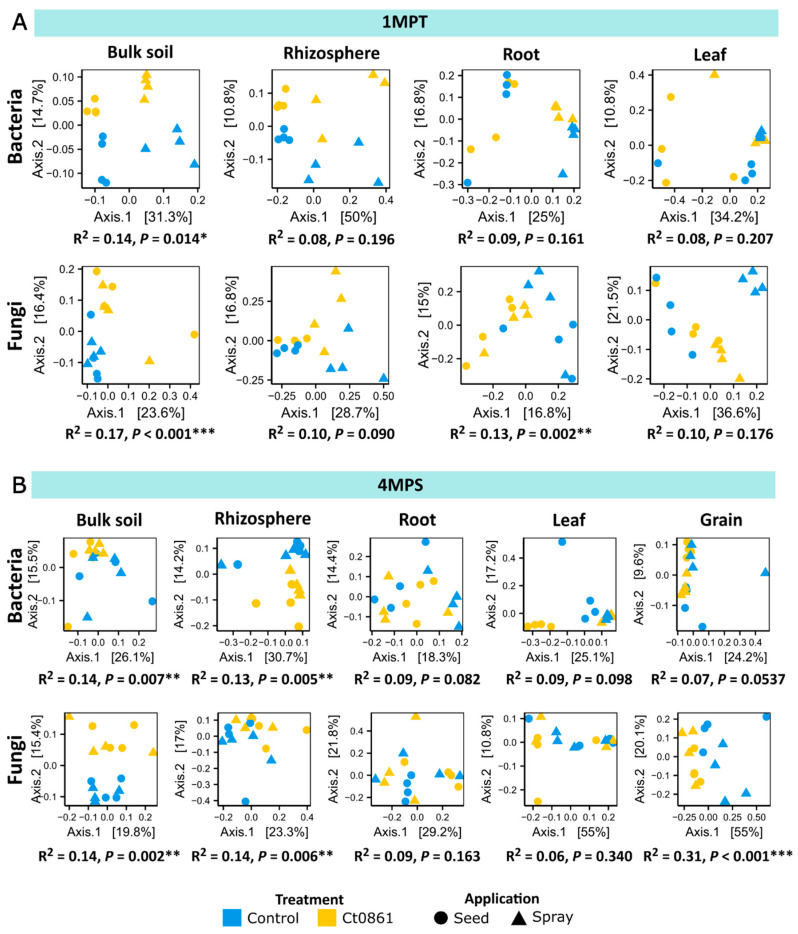
Beta-diversity patterns of the maize-associated microbiome of plants treated or not with Ct0861. Principal coordinate analysis (PCoA) based on Bray–Curtis dissimilarity of bacterial and fungal communities in each compartment unconstrained by “treatment” and “application” at sampling time point: (**A**) 1MPT (one month post-treatment) and (**B**) 4MPS (four months post-sowing). The relative contribution (R^2^) of the factor “treatment” on community dissimilarity and its significance were tested with PERMANOVA (*n* = 16). (*p* < 0.05 *, *p* < 0.01 **, *p* < 0.001 ***).

**Figure 4 plants-14-03236-f004:**
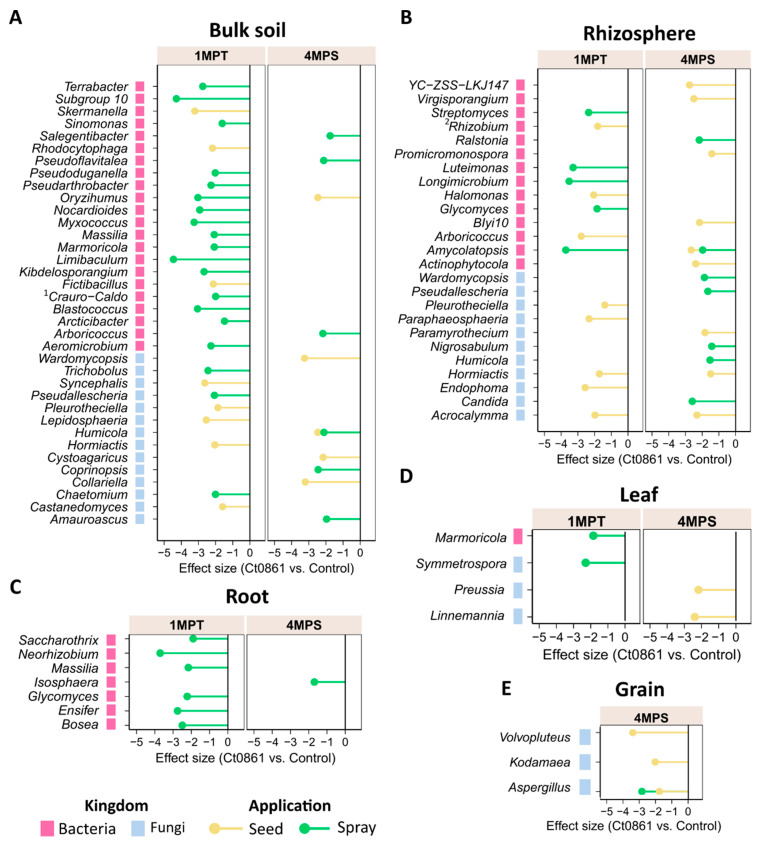
Differential abundant bacterial and fungal taxa in Ct0861-treated plants at different compartments, sampling time points and application methods. Bars represent effect size, which indicates the magnitude of difference in relative abundance between Ct0861-treated and control groups, standardized by the variability across Monte Carlo instances, as calculated by ALDEx2. Shown are bacterial and fungal taxa that were significantly (*p* < 0.05) differentially abundant in (**A**) bulk soils, (**B**) rhizospheres, (**C**) roots, (**D**) leaves, and (**E**) grains of Ct0861-treated maize plants relative to control plants. ^1^
*Craurococcus-Caldovatus.*
^2^
*Allorhizobium-Neorhizobium-Pararhizobium-Rhizobium*.

**Figure 5 plants-14-03236-f005:**
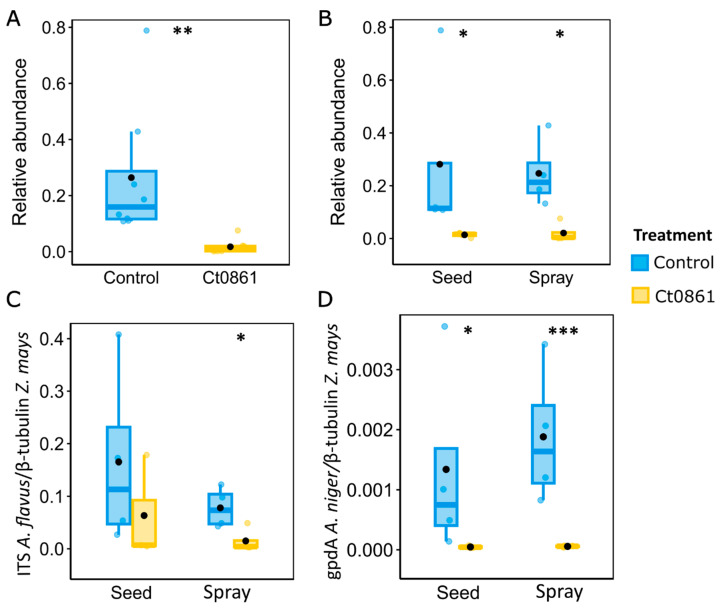
Quantification of *Aspergillus* spp. in the grains of maize plants treated or not with Ct0861. (**A**) Relative abundance of *Aspergillus* spp. in the aggregated data set of grains of maize plants treated or not with Ct0861. (**B**) Relative abundance of *Aspergillus* spp. in grains of maize plants treated or not with Ct0861 either as seed or spray application. Quantification by qPCR of (**C**) *A. flavus* and (**D**) *A. niger* biomass in grain samples of maize plants treated or not with Ct0861 either on the seed before sowing or as a spray at one month after sowing. Internal line in box-plot boxes indicates the median or second quartile (Q2), upper line of the boxes indicates the third quartile (Q3) and lower line the first quartile (Q1) of the data. Mean is represented by a black dot. Asterisks denote statistically significant differences in Student’s *t*-test (*p* < 0.05 *, *p* < 0.01 **, *p* < 0.001 ***).

**Figure 6 plants-14-03236-f006:**
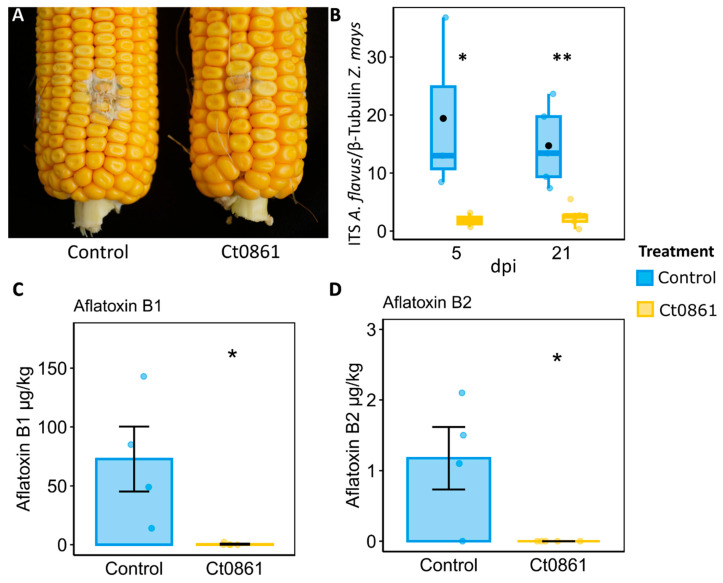
Protective effect of Ct0861 treatment on maize grains infected with *A. flavus*. (**A**) Representative image showing the mycelial growth of *A. flavus* at 21 dpi on the inoculation site in cobs of maize plants inoculated or not with Ct0861. (**B**) Quantification of *A. flavus* biomass in grains of maize plants inoculated or not with Ct0861 at five and 21 dpi with *A. flavus* strain NRRL 6540. Aflatoxin B1 (**C**) and B2 (**D**) concentration (µg/kg) in grains of maize plants inoculated or not with Ct0861 at 21 dpi with *A. flavus* strain NRRL 6540. Internal line in box-plot boxes indicates the median or second quartile (Q2), upper line of the boxes indicates the third quartile (Q3) and lower line the first quartile (Q1) of the data. Mean is represented by a black dot. Asterisks denote statistically significant differences in Student’s *t*-test (*p* < 0.05 *, *p* < 0.01 **).

**Table 1 plants-14-03236-t001:** Primer sequences used for amplicon sequencing.

Target Region	Primer ID	Sequence (5′-3′)	Annealing T (°C)	Reference
16S-V4	515f	GTGCCAGCMGCCGCGGTAA	52	[34]
	806r	GGACTACHVGGGTWTCTAAT		
ITS1	ITS1-F	CTTGGTCATTTAGAGGAAGTAA	50	[35]
	ITS2	GCTGCGTTCTTCATCGATGC		[36]

**Table 2 plants-14-03236-t002:** Primer sequences used for fungal biomass quantification by real-time qPCR.

Fungal Species	Primers ID	Forward Sequence	Reverse Sequence	Annealing T (°C)	Ref.
*C. tofieldiae*	PRB110_00602#3	CTCGTGTGACTGCGTTGTTG	TGGGTTGTGCGGGATTCAG	60	[26]
*A. flavus*	Af2	ATCATTACCGAGTGTAGGGTTCCT	GCCGAAGCAACTAAGGTACAGTAAA	60	[51]
*A. niger*	gpdA	TCAAACCGACATTGCGAGAA	TCGCCGTGGTTGATGCT	60	[52]

**Table 3 plants-14-03236-t003:** Effect of Ct0861 seed^1^ or spray application on growth and yield parameters of maize plants grown in an open field.

Treatments	Stem Height (m)	Shoot Fresh Weight (g)	Root Dry Weight (g)	No. of Cobs per Plant	Cob Weight per Plant (g)	1000 Grains Weight (g)	Yield per Plant (g)	Yield (t/ha)
Seed application ^1^								
Control	2.47 ± 0.24 b	619 ± 124 a	16.6 ± 5 a	1.25 ± 0.45 a	198.2 ± 30 a	376 ± 22 a	169 ± 28.8 b	12.5 ± 2.1 b
Ct0861	2.63 ± 0.28 a	634 ± 139 a	18.5 ± 7 a	1.31 ± 0.5 a	237.3 ± 30.4 a	399 ± 3 a	206.8 ± 23.3 a	15.2 ± 1.7 a
Spray application								
Control	2.46 ± 0.24 b	580 ± 153 b	14.5 ± 6 b	1.19 ± 0.4 a	193.8 ± 12.7 b	371 ± 5 b	170 ± 26.6 a	12.5 ± 2 a
Ct0861	2.7 ± 0.24 a	728 ± 144 a	25.5 ± 10 a	1.38 ± 0.5 a	235.1 ± 25.2 a	393 ± 8 a	204.5 ± 25.4 a	15.1 ± 1.9 a

Data show means ± standard deviations. Stem height (*n* = 80), root dry weight and number of cobs per plant (*n* = 16) were analyzed by Two-Sample Wilcoxon-Mann–Whitney Test. Shoot fresh weight (*n* = 16), cob weight per plant (*n* = 4), 1000 grains weight (*n* = 8), yield per plant (*n* = 4) and yield (*n* = 4) were analyzed by Welch Two-Sample *t*-test. Different letters within each column indicate significant differences among treatments (*p* < 0.05). All statistical analyses were conducted between control and Ct0861 groups within each application method. ^1^ Data of maize growth and yield for seed application were already published in reference [26].

## Data Availability

The datasets supporting the conclusions of this article are available in the National Center For Biotechnology Information (NCBI) repository under the accession numbers PRJNA1056947 (control samples) and PRJNA1209672 (Ct0861 samples).

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
