# Peer review of "Microbiome Analysis Reveals Biocontrol of Aspergillus and Mycotoxin Mitigation in Maize by the Growth-Promoting Fungal Endophyte Colletotrichum tofieldiae Ct0861"

_plants, 2025, doi:10.3390/plants14213236_

Round 1

Reviewer 1 Report

Comments and Suggestions for Authors

In this manuscript, Sandra Díaz-González and colleagues evaluated the potential of Colletotrichum tofieldiae strain Ct0861 as a bioinoculant and its impact on maize-associated bacterial and fungal microbiomes. . I have following comments:

1, For the Abstract section, more data should be presented. For instance, authors stated that “Both microbiome data and qPCR quantification confirmed a significant reduction in Aspergillus spp. abundance in Ct0861-treated plants.”, but no data was provided.

2, For the introduction section, more information should be provided. For instance, plant species in the citation 17 should be included.

3, Methods for random sampling should be introduced in the Methods.

4, For the results, the difference between one or two asterisks in Figures 1,4 and 5 should be explained in the legend.

5, For the discussion, I would like to see the discussion section was divided into subsections with appropriate titles.

6, A Figure depicting the working model of Colletotrichum tofieldiae strain Ct0861 in biocontrol of Aspergillus and mycotoxin mitigation in maize is appealing.

Author Response

In this manuscript, Sandra Díaz-González and colleagues evaluated the potential of Colletotrichum tofieldiae strain Ct0861 as a bioinoculant and its impact on maize-associated bacterial and fungal microbiomes. I have following comments:

We sincerely thank the reviewer for taking the time to carefully read and evaluate our manuscript. We greatly appreciate the constructive comments and suggestions, which have substantially improved the clarity and overall quality of our work. In response to your and other reviewers’ feedback, we have made several major revisions throughout the manuscript. Specifically:

  • We have incorporated the former Supplementary Figure S1B as a main figure (now Figure 1) and renumbered all figures accordingly, while Supplementary Figure S1A remains as Figure S1.
  • We have moved the soil analysis table from the Supplementary Methods to a Supplementary Table (now Table S1) and adjusted all supplementary table numbers and in-text references.
  • We have removed the qPCR primer table from the Supplementary Methods and created two new tables in the main text: one for amplicon sequencing primers (Section 2.6) and another for fungal biomass quantification by qPCR (Section 2.9), now Tables 1 and 2, respectively. The previous Table 1 is now Table 3.
  • We have reorganized the Discussion section into subsections with appropriate titles to improve readability and structure.

Below, we address each point raised by the reviewer in detail (line numbers refer to the accepted changes file):

1, For the Abstract section, more data should be presented. For instance, authors stated that “Both microbiome data and qPCR quantification confirmed a significant reduction in Aspergillus spp. abundance in Ct0861-treated plants.”, but no data was provided.

We thank the reviewer for this comment. We fully agree that presenting supporting data is essential; however, we believe that the Abstract section should serve primarily as a concise summary of the study rather than a detailed presentation of numerical results. In addition, the journal limits the Abstract to 200 words, which constrains the inclusion of detailed data. All the specific data and figures supporting the statement regarding the reduction of Aspergillus spp. abundance in Ct0861-treated plants are provided and discussed in detail in the main text and figures. For this reason, we respectfully consider that the current level of detail in the Abstract is appropriate.

2, For the introduction section, more information should be provided. For instance, plant species in the citation 17 should be included.

We thank the reviewer for this helpful suggestion. We have revised the sentence in the Introduction to include specific examples of plant species that benefit from Serendipita indica colonization, based on the cited reference (Saleem et al., 2022). The sentence now reads as follows (lines 53-57):

“A well-known example is Serendipita indica, which promotes vegetative growth, nutrient uptake and plant yield across multiple hosts, including Arabidopsis thaliana, barley (Hordeum vulgare), maize (Zea mays), wheat (Triticum aestivum), tomato (Solanum lycopersicum), and cucumber (Cucumis sativus), while also improving tolerance to biotic and abiotic stressors [17].”

3, Methods for random sampling should be introduced in the Methods.

We thank the reviewer for this helpful comment. The methods used for random sampling were already described in the original manuscript; however, to improve clarity we have now expanded this section. Specifically, we added the following sentences:

Lines 156-159: “Randomization was achieved by assigning each plant a unique identifier, followed by using the formula RANDBETWEEN in Microsoft Excel as a random number generator to select plants for harvesting.”

Line 166: “One maize plant of the central row of each plot (4 plants per treatment, 16 plants in total, Figure 1) was randomly selected. Random selection was done as described in section 2.4.”

We believe this addition clarifies our random sampling procedure as requested.

4, For the results, the difference between one or two asterisks in Figures 1,4 and 5 should be explained in the legend.

We thank the reviewer for pointing this out. We have now added an explicit explanation of the meaning of the asterisks in the figure legends. Specifically, the legends for Figures 2, 5 and 6 (former Figures 1, 4 and 5) now include: “*P < 0.05; **P < 0.01; ***P < 0.001.”

5, For the discussion, I would like to see the discussion section was divided into subsections with appropriate titles.

We thank the reviewer for this constructive suggestion. In response, we have reorganized the Discussion section into subsections with appropriate titles to improve clarity and readability. We believe this revised structure will help readers follow the flow of arguments more easily and appreciate the key points of our study.

6, A Figure depicting the working model of Colletotrichum tofieldiae strain Ct0861 in biocontrol of Aspergillus and mycotoxin mitigation in maize is appealing.

We thank the reviewer for this interesting suggestion. While a schematic figure depicting a working model of Ct0861 in the biocontrol of Aspergillus and mycotoxin mitigation in maize would indeed be appealing, we believe that including such a model at this stage would be premature. At present, we do not have direct evidence of the underlying mechanism. Our data suggest that the effect is unlikely to be due to direct antagonism, as (i) Ct0861 was not detected in maize grain and (ii) antibiogram assays showed no antagonistic activity between Ct0861 and Aspergillus. These observations point towards a possible induced-resistance effect; however, without direct evidence, we consider it inappropriate to present a hypothetical model in this manuscript. We believe that such a figure would be better suited for future publications once the mechanism has been elucidated.

Reviewer 2 Report

Comments and Suggestions for Authors

The manuscript focuses on biological control studies involving a maize fungal endophyte.  It was a pleasure reading a well-coordinated research.

Some comments:

The manuscript includes a substantial number of Supplementary files (e.g., Supplementary Table S1, Figure S1); however, some of these are not referenced in the text. Omit the word supplementary when it is referenced in the text, as Table S1 means Supplementary Table 1, and so on.

Figure SA1 is needed in Section 2.3.

Supplementary Methods file: it contains soil analysis methods and results as well. This file does not have proper code, but it should be Table S1 for Section 2.3. However, the data are not discussed in the text; therefore, their supplementation is questionable.

There is a Figure S1B, which is very good for Section 2.5. I suggest using it as an in-text Figure instead of a Supplement.

In Section 2.6, there is no reference to Supplement Methods Table 4. That should be Table S1.

Now it is very disturbing to find the information. The Table of the PCR probes and at least the annealing T should be in the main text, in Sections 2.6 and 2.9

In Section 2.10, several mycotoxins are mentioned as quantified compounds from the grains; however, most of them are not presented or discussed later, nor is there a supplementary data table.

Were the inoculants (both) tested for secondary metabolite production? With the LC-MS/MS technique, you have the possibility.

In Table 1, was the T-test was done only between control and seed or spray-treated plants separately? The letters show the relation only between the treatment and its control, or are both treatments being compared? I suggest explaining it, and also checking the letters of the differences. The head of the table: complicated to understand as the lines are slipped to each other.

At Page 8, there are problems in the text: „Error! Reference source not found”

In Section 3.2, Figure S2 is needed.

In Figure 3, what is the effect size? Is it a fold change?

In Figure 4, is it normal that the data points are not on the same vertical line?

On page 16, what is called „severe” microbiome disruption” or „strong microbiome alteration”?

On Page 17: „microbial IR”- Microbially induced resistance

„IR can be mediated by salicylic acid”- do you mean SAR?

Interesting that in Arabidopsis, CT0861 activates both SAR and ISR. Is it right? Is it working in the same way in maize?

The section „Supplementary material” does not contain all the supplements.

In References, the DOI is missing in several cases, and also, text repetitions are present, e.g., at 84.

Author Response

The manuscript focuses on biological control studies involving a maize fungal endophyte. It was a pleasure reading a well-coordinated research. Some comments:

We sincerely thank the reviewer for the positive feedback and for taking the time to carefully read our manuscript. We are pleased to know that the study was perceived as well-coordinated and engaging. In response to your and other reviewers’ feedback, we have made several major revisions throughout the manuscript. Specifically:

  • We have incorporated the former Supplementary Figure S1B as a main figure (now Figure 1) and renumbered all figures accordingly, while Supplementary Figure S1A remains as Figure S1.
  • We have moved the soil analysis table from the Supplementary Methods to a Supplementary Table (now Table S1) and adjusted all supplementary table numbers and in-text references.
  • We have removed the qPCR primer table from the Supplementary Methods and created two new tables in the main text: one for amplicon sequencing primers (Section 2.6) and another for fungal biomass quantification by qPCR (Section 2.9), now Tables 1 and 2, respectively. The previous Table 1 is now Table 3.
  • We have reorganized the Discussion section into subsections with appropriate titles to improve readability and structure.

Below, we address each point raised by the reviewer in detail (line numbers refer to the accepted changes file):

The manuscript includes a substantial number of Supplementary files (e.g., Supplementary Table S1, Figure S1); however, some of these are not referenced in the text. Omit the word supplementary when it is referenced in the text, as Table S1 means Supplementary Table 1, and so on.

We thank the reviewer for this comment. We have carefully checked the manuscript to ensure that all supplementary materials are properly referenced in the text. Following the reviewer’s suggestion, we have omitted the word “Supplementary” when citing them (e.g., “Table S1” instead of “Supplementary Table S1”) to maintain consistency.

Figure SA1 is needed in Section 2.3.

We thank the reviewer for this suggestion. We have now added the reference to Figure S1 (former Figure S1A) in Section 2.3. The sentence now reads: “The study was conducted from May to October 2018 in the field of an experimental station located at Pilar de la Horadada, Alicante (37°51'46.0"N–0°48'30.6"W, 35 m above mean sea level) in the south-eastern part of the Iberian Peninsula, as described in [26] and [28] (Figure S1)” (lines 113-116).

Supplementary Methods file: it contains soil analysis methods and results as well. This file does not have proper code, but it should be Table S1 for Section 2.3. However, the data are not discussed in the text; therefore, their supplementation is questionable.

We thank the reviewer for this observation. Our intention in including the soil analysis results was to demonstrate that the physical and chemical properties of the soils in Ct0861-treated and control plots were comparable at the start and end of the experiment, ensuring homogeneity across treatments. To clarify this, we have now moved the data to Table S1 and cited it directly in Section 2.3. We have also added a brief statement in the text indicating that no differences were found between the soil of the treatments. Now it reads (lines 136-139):

“Soil physical and chemical properties were analysed by Fitosoil Laboratorios SL in both control and Ct0861-treated plots to ensure soil homogeneity across treatments (Table S1). No differences were found among the plots of the different treatments either before or after the experiment.”

Accordingly, we have renumbered the rest of Supplementary Tables.

There is a Figure S1B, which is very good for Section 2.5. I suggest using it as an in-text Figure instead of a Supplement.

We thank the reviewer for this helpful suggestion. Following it, we have moved the former Supplementary Figure S1B into the main text as Figure 1 and updated all figure numbers and corresponding references throughout the manuscript accordingly. As commented above, the former Supplementary Figure S1A remains as Supplementary Figure S1.

In Section 2.6, there is no reference to Supplement Methods Table 4. That should be Table S1. Now it is very disturbing to find the information. The Table of the PCR probes and at least the annealing T should be in the main text, in Sections 2.6 and 2.9

We appreciate the reviewer’s helpful comment. Following this suggestion, we have removed the qPCR primer table from the Supplementary Methods and created two new tables in the main text: one listing the amplicon sequencing primers (Table 1, Section 2.6) and another listing the primers used for fungal biomass quantification by qPCR (Table 2, Section 2.9). We have also added the annealing temperatures for each primer pair, as requested. All tables and in-text references have been renumbered accordingly.

In Section 2.10, several mycotoxins are mentioned as quantified compounds from the grains; however, most of them are not presented or discussed later, nor is there a supplementary data table.

We thank the reviewer for this comment. While several mycotoxins were analyzed, as described in Material and Methods section 2.10, only aflatoxins B1 and B2 were detected in the grain samples. This is stated in the Results section (lines 538-540): “The levels of various mycotoxins were measured in grains collected 21 dpi with A. flavus. Among the analyzed mycotoxins, only aflatoxins B1 and B2 were detected in the samples.” These data are presented in Figures 6C and 6D (former Figure 5C and D) and are discussed accordingly. Since the other mycotoxins were not detected and A. flavus is known to produce only aflatoxins, we consider it unnecessary to include additional discussion or supplementary tables for the non-detected compounds.

Were the inoculants (both) tested for secondary metabolite production? With the LC-MS/MS technique, you have the possibility.

We thank the reviewer for this suggestion. The A. flavus strain used was described as toxigenic through aflatoxin production (Wicklow et al., 1981), and we have confirmed its in planta production (in maize grains) through LC-MS/MS (see material and methods section 2.10). Indeed, analyzing secondary metabolite production in axenic cultures using LC-MS/MS would also be interesting, but we believe that this is beyond the scope of the current study.

Wicklow, D.T.; Shotwell, O.L.; Adams, G.L. Use of Aflatoxin-Producing Ability Medium to Distinguish Aflatoxin-Producing Strains of Aspergillus Flavus. Appl Environ Microbiol 1981, 41, 697–699, doi:10.1128/AEM.41.3.697-699.1981

In Table 1, was the T-test was done only between control and seed or spray-treated plants separately? The letters show the relation only between the treatment and its control, or are both treatments being compared? I suggest explaining it, and also checking the letters of the differences. The head of the table: complicated to understand as the lines are slipped to each other.

We thank the reviewer for this comment. We have revised Table 3 (former Table 1) to improve readability and added a detailed footnote to clarify the statistical analyses. The footnote now reads (lines 345-351:

“Data show means ± standard deviations. Stem height (n=80), root dry weight and number of cobs per plant (n=16) were analyzed by Two Sample Wilcoxon-Mann-Whitney Test. Shoot fresh weight (n=16), cob weight per plant (n=4), 1000 grains weight (n=8), yield per plant (n=4) and yield (n=4) were analyzed by Welch Two-Sample t-test. Different letters within each column indicate significant differences among treatments (P < 0.05). All statistical analyses were conducted between control and Ct0861 groups within each application method. Data of maize growth and yield for seed application were already published in reference [26].”

At Page 8, there are problems in the text: „Error! Reference source not found”

We thank the reviewer for noticing this error. The reference has now been corrected and properly points to Table 3 (former Table 1, lines 356-365).

In Section 3.2, Figure S2 is needed.

We thank the reviewer for this comment. The reference has been updated in the text: “Supplementary Figure S2” was changed to “Figure S2” to ensure consistency and clarity. (lines 390 and 401).

In Figure 3, what is the effect size? Is it a fold change?

We thank the reviewer for this question. The “effect size” in Figure 4 (former Figure 3) refers to the output of the ALDEx2 analysis, which measures the magnitude of difference in relative abundance of each feature between the two conditions. This value is standardized by the variability across Monte Carlo instances and is not a direct fold change. Positive values indicate higher abundance in the Ct0861-treated group, while negative values indicate higher abundance in the control group. To clarify this, we have reformulated the figure caption. It now reads as follows (lines 474-480):

“Figure 4. Differential abundant bacterial and fungal taxa in Ct0861-treated plants at different compartments, sampling time points and application methods. Bars represent effect size, which indicates the magnitude of difference in relative abundance between Ct0861-treated and control groups, standardized by the variability across Monte Carlo instances, as calculated by ALDEx2. Shown are bacterial and fungal taxa that were significantly (P < 0.05) differentially abundant in (A) bulk soils, (B) rhizospheres, (C) roots, (D) leaves, and (E) grains of Ct0861-treated maize plants relative to control plants.”

In Figure 4, is it normal that the data points are not on the same vertical line?

We thank the reviewer for this observation. The data points in Figure 5 (former Figure 4) are intentionally slightly offset horizontally because the plot includes a jitter layer over the boxplots. This is a common visualization technique in ggplot2 (RStudio) to prevent overlapping points and allow all individual values to be visible. While the points are not exactly on the same vertical line, their vertical position accurately represents the measured values.

On page 16, what is called „severe” microbiome disruption” or „strong microbiome alteration”?

We thank the reviewer for this comment. In the sentences referred to, terms such as “severe microbiome disruption” or “strong microbiome alteration” are intended to convey that Ct0861 treatment did not cause major or drastic changes in the overall microbiome structure or composition. Instead, the observed effects on microbial communities were subtle yet meaningful, reflecting nuanced shifts in diversity and composition rather than large-scale disturbances.

On Page 17: „microbial IR”- Microbially induced resistance

We thank the reviewer for this comment. We have changed “microbial” by “microbially” (line 662)

„IR can be mediated by salicylic acid”- do you mean SAR?

We thank the reviewer for this comment. We used the term “induced resistance” (IR) following the recommendations of De Kesel et al. (2021), where IR is employed as a general umbrella term encompassing all forms of defense priming or stimulation induced by microbes or other stimuli. While SAR (systemic acquired resistance) is classically defined as SA-dependent and triggered by necrotizing pathogens, and ISR (induced systemic resistance) is typically JA/ET-mediated and triggered by PGPR or PGPF, we mentioned that “IR can be mediated by salicylic acid (SA)” to indicate that some forms of microbially induced resistance may involve SA-dependent pathways, including accumulation of pathogenesis-related (PR) proteins. Therefore, based on the recommendations of De Kesel et al., we consider IR the most appropriate term to describe the indirect defense responses that may be activated by Ct0861.

De Kesel, J., Conrath, U., Flors, V., Luna, E., Mageroy, M. H., Mauch-Mani, B., ... & Kyndt, T. (2021). The induced resistance lexicon: do’s and don’ts. Trends in plant science, 26(7), 685-691.

Interesting that in Arabidopsis, CT0861 activates both SAR and ISR. Is it right? Is it working in the same way in maize?

We thank the reviewer for this interesting observation. Indeed, in Arabidopsis Ct0861 has been shown to activate SA, ET and JA-associated pathways. However, whether Ct0861 triggers similar responses in maize remains to be determined. Our current study focused on plant performance and microbiome composition; we are now exploring the underlying defense mechanisms in maize in ongoing experiments.

The section „Supplementary material” does not contain all the supplements.

We thank the reviewer for noting this. We have carefully revised the Supplementary Material section to ensure that all supplementary figures, tables, and files are now correctly listed and numbered. The section has been updated to include all supplementary items referenced in the main text (lines 710-724).

In References, the DOI is missing in several cases, and also, text repetitions are present, e.g., at 84.

We thank the reviewer for pointing this out. We have carefully revised the reference list and made the corrections needed.

Round 2

Reviewer 1 Report

Comments and Suggestions for Authors

Authors have addressed my concerns in the revision.

Reviewer 2 Report

Comments and Suggestions for Authors

I endorse the publication.